# One-Part Plastic Formable Inorganic Coating Obtain from Alkali-Activated Slag /Starch(CMS) Hybrid Composites

**DOI:** 10.3390/molecules25040844

**Published:** 2020-02-14

**Authors:** Xuesen Lv, Yao Qin, Zhaoxu Lin, Zhenkun Tian, Xuemin Cui

**Affiliations:** School of Chemistry and Chemical Engineering and Guangxi Key Lab of Petrochemical Resource Processing and Process Intensification Technology, Guangxi University, Nanning 530004, Chinaw1638876932@163.com (Y.Q.); Linzhaoxu052152905@163.com (Z.L.); tzk1240127171@163.com (Z.T.)

**Keywords:** AAM, Coating, Carboxymethyl starch, Rheology, Slag

## Abstract

Coating technology can be applied to decorate building constructions. Alkali-activated materials (AAM) are promising green and durable inorganic binders which show potential for development as innovative coating. In the paper, the possibility of using AAM composited with starch (CMS) as a novel plastic formable inorganic coating for decorating in building was investigated. The rheological properties, including plastic viscosity, yield stress, and thixotropy were considered to be critical properties to obtain the working requirements. Four different mixtures were systematically investigated to obtain the optimum formulation, and then were used to study their hardened properties, such as mechanical strengths (compressive, flexural, and adhesive strength), drying shrinkage, cracking behavior, and microstructure. Study results found that CMS could quickly and efficiently be hydrolyzed in an alkaline solution to produce organic plastic gel which filled in AAM paste, leading to the significant improvement of coating consistency, plastic viscosity, and thixotropy. The optimum coating composited with 15.40 wt% CMS shows a relatively stable rheological development, the setting time sufficient at higher than 4 h. Furthermore, CMS shows a significant positive effect on the cracking and shrinkage control due to padding effect and water retention of CMS, which results in no visible cracks on the coating surface. Although the mechanical strength development is relatively lower than that of plain AAM, its value, adhesive strength 2.11 MPa, compressive strength 55.09 MPa, and flexural strength 8.06 MPa highly meet the requirements of a relevant standard.

## 1. Introduction

Coating technology can be applied to building and construction surface for decorating, protecting, and functionalizing, which plays a critical role in beautifying urban environments and city development [1,2,3,4]. In general, coatings can be classified as inorganic or organic coatings [4,5,6,7]. Inorganic coatings are commonly obtained from cross-linking binders such as silicon resins, silica sol, water and solvent silicates, and alkaline metal silicate solutions, etc. Compared to organic coatings, inorganic coatings show much advantage in excellent aging resistance, good chemical-physical attack resistance, zero volatile organic compounds (VOCs, toxic gases) emissions, and low materials and processing cost [2,7,8,9].

At present, traditional inorganic coatings are mainly manufactured from silicate solutions. In fact, until now, the real application of inorganic coatings used for decorating in building is very limited due to their inherent weakness. The main factors influencing the development of silicate coatings are the poor water resistance caused by water soluble M_2_O (*M* = Na or K) [2,8,10,11], as well as they are easily cracking during solidifying due to their brittleness [8,12]. Moreover, silicate coatings usually show inadequate workability properties in practical engineering, resulting from their rheological performance being very susceptible to pH, temperature, and humidity fluctuation [13,14,15].

Alkali-activated materials (AAM) are the novel inorganic binders, which are usually produced by mixing aluminosilicates with an alkaline solution. This process produces eco-efficient products that allow industrial wastes to be effectively utilized without causing more harm to the environment. Ground granulated blast furnace slag (GGBS) is one of such waste residues from the iron industry that can be alkali-activated due to its high fraction amorphous phase. Thus, the application of alkali-activated slag presents outstanding advantage in simple manufacture process, low-price, easy accessibility of raw materials, energy conservation, low-carbon emission, etc. AAM binders have nanostructural amorphous or poorly-crystalline gel of N-A-S(-H)/C-(N)-A-S-H, which results in outstanding characteristics in high early mechanical strength, excellent physical-chemical stability, great corrosion resistance, good workability, and adhesive strength between different materials [9,16,17,18,19]. Moreover, compared to the smooth thin film produced by the traditional coatings, fresh AAM is a slurry-like which easily realizes the desired artistic shape as the coating layer is thicker. This is the fundamental advantage for AAM-based decorative coating. In this regard, AAM has become a novel candidate for developing inorganic coatings for architectural decorating.

The construction process of decorative coatings requires good plastic formable capacity ensuring the coating is flexibly formed into a complex layer appearance so that it meets people’s aesthetic expectations, which highly relates to the plastic viscosity and thixotropy. Few papers have investigated the relationship between different alkaline activators and solid particles on the rheological behavior of AAM [20,21,22], but studies have found that AAM often shows the excellent fluidity due to the high electrostatic repulsion of the particle surface which was caused by the fast diffusion rate of the Al and Si elements in high content [OH]^−^ [22,23]. Furthermore, there is no colloidal interaction or flocculation structure between AAM particles, and lots of residua water is present in aluminosilicates particles in a freely moving status which does not take part in reaction [20,23]. As a result, fresh AAM commonly shows poor cohesiveness and thus cannot directly be used as a coating. This issue can be overcome by compositing organic modifiers, using entangled organic molecular long chains to fill or intertwine between particle gaps so as to improve cohesiveness. Unfortunately, many studies have found that organic modifiers’ molecular structure or their functional groups will be destroyed in high alkalinity due to the poor chemical compatibility problem [19,24]. It is still, to this day, a challenge to find useful organic modifiers or a simple method by which to effectively change AAM visco-elastic behavior so that it meets engineering requirements.

The objective of this paper was to develop a one-part plastic formable AAM coating for building decorating by using starch (CMS) organic/inorganic hybrid composite. The coating performances were systematically studied, including rheology behavior, cracking behavior, mechanical strength, and microstructure. Such a coating prepared by using powder technology just needs to be mixed with water prior to use; CMS and the AAM precursors can rapidly dissolve in cold water and finally create a three-dimensional AAM/CMS network. The coating manufactured by optimal formulation possesses excellent workability, showing no cracks and high mechanical strengths. This novel coating shows extensive developing potential for commercial application and academic research.

## 2. Results and Discussion

### 2.1. Rheology and Workability

The effect of CMS on the flow behavior of fresh AAM is shown in Figure 1. All samples rather obviously showed shear thinning behavior. They also revealed that the pure AAM paste showed excellent fluidity while it did not possess thixotropy behavior (Figure 1A), but it appeared that CMS had a strong enhancement on thixotropy and decrement on fluidity (Figure 1B–D). Rheological behavior of fresh paste originated from flocculation caused by Van Der Waals force, hydrogen bond, electrostatic forces, colloidal and contact interactions between particles [25,26]. Few papers have investigated the basic AAM rheology behavior related to alkali-activated reaction. It was found that the excellent fluidity of AAM primarily results from the high electrostatic repulsion of particles surface caused by the fast diffusion rate of Al and Si in high content alkalinity. [14,22]. Moreover, there is no colloidal interaction or flocculation structure between fresh AAM particles, and lots of residual water actually exists as freely moving status which do not take part in the reaction, resulting in poor cohesiveness and viscosity of AAM due to the water dilution effect [23]. In this study, the thixotropy strongly increased after the addition of CMS, implying the improvement of cohesiveness. This result might be caused by the colloidal interaction of CMS gel between slag particles. On the other hand, the multiple hydrophilic group (-CH2COONa, -OH) in the glucopyranose unit of CMS fixed part of the residual water by hydrogen bond, which relatively reduced the freely moving water in the suspension. Such results indicate that CMS strongly improved the formable ability of AAM. CMS can be used as a modifier to effectively change AAM visco-elastic behavior so that it meets the engineering requirements as coating. 

Figure 2 shows the effect of CMS on the improvement degree of initial plastic viscosity, yield stress, and thixotropy. It can be seen that the improvement of initial plastic viscosity (Figure 2A) and initial thixotropy (Figure 2C) strongly improved with the increase of CMS, suggesting that CMS significantly improved the plasticity of AAM paste due to the product of resilient intertwining chains of CMS, and therefore leading to the plastic formable ability. In addition, CMS possessed a significant effect on altering viscosity and thixotropy of AAM pastes in a large numeric range, indicating that CMS was compatible and efficient with AAM. The initial yield stress also improved with the increase of CMS, suggesting that the mechanical stability of the AAM pastes were enhanced. Although the yield stress did not significantly improve like the viscosity and thixotropy, CMS showed a high enough yield stress that could offer high enough cohesiveness to the coating structure with good flowability, whereas a sudden high initial yield stress of 566 Pa was obtained when composited 15 g CMS, corresponding to the stiffing problem. The reasons probably resulted from the unsuitable water/solid content, and CMS exhibited excellent water binding ability which resulted from the hydrogen effect to fix water molecules; thus, the available water needed for dissolving and wetting AAM precursors was relatively reduced, resulting in stiff behavior. It should be noted that too-low yield stress of coating may bring about sagging or even water/solid separating due to poor cohesiveness, and while too-high yield stress may cause stiffing, the coating could not be freely coated or brushed into a continual film.

As AAM is a high reactive system and easily solidified, the evolution of coating flow behavior with time is one of the very important properties which relates to the operable time and workability during real application [22,23,27]. The rheological evolution of coating with setting time was investigated, the rheological parameters including yield stress, plastic viscosity, and thixotropy are pasted in Figure 3. Though the rheological behavior of sample #1 was very stable with time evolution, the extremely low yield stress and insufficient thixotropy would highly respond to the poor cohesiveness and segregation resistance, leading easily to the coating structure sagging and segregating. Therefore, plain AAM cannot totally meet the working requirement as coating. However, a reverse phenomenon was observed when the samples composited with CMS. In general, the plastic viscosity, yield stress and thixotropy increased with time due to co-polymerization. However, the rheological parameters of sample 4# shows a large change with time, suggesting that the unstable workability meanwhile too stiff even with a high yield stress, leading to the uncontrollable workability problem during practical application. In comparison, the rheological parameters between sample 2# and 3# showed a high enough initial thixotropy and plastic viscosity of 26520 Pa/s and 9 Pa s respectively, and the yield stress less than 389 Pa even after 4 h reaction. This stable workability and rheological performances could enable the coating to be easily and freely formed into various artistic patterns with complex layer structure. Therefore, sample 2# and 3# were identified as the optimum in term of workability here.

### 2.2. Mechanical Strength

The compressive, flexural, and adhesive strengths of AAM samples with different CMS and curing time are shown in Figure 4. In general, the mechanical strengths all decreased with the increasing content of CMS but increased with curing time. This result was contrary to other previous studies that used organic resins and polymer fibers to reinforce AAM [28,29,30]. For the result in this study, it can be explained for that CMS diluted the AAM binder content. Moreover, the mechanical strength of CMS binder structure was much poorer than that of AAM. Nevertheless, according to the Chinese inorganic architectural coating standard JC/T 423 91, the adhesive strength should not be less than only 1.0 MPa. In this study, a high adhesive strength of 2.1 MPa was obtained even composited with a high content of 15.4 wt% CMS. Meanwhile, the 28 d compressive strength and flexural strength was higher than 55 MPa and 8.0 Mpa, respectively, which closed to the strength of Ordinary Portland Cement 525. Based on the experimental results, it can be concluded that the strengths satisfied the technology requirements well, suggesting these strengths can provide adequate hardness for AAM coating.

Adhesive strength is a very important parameter for coating. In order to investigate the adhesive characteristics between cement mortar substrate and the prepared coating, 28 d curing sample was vertically split in Figure 5 for bonding area observation. It revealed the dense microstructure of AAM coating and interfacial bonding zone, which led to the high adhesive strength of coating due to the typical AAM C-A-S-H/A-S-H gels chemically bonded into the cement mortar substrate. However, lots of smashed fragments presented in sample #1 (Figure 5A) after being split, which resulted from typical brittle behavior with low ductility and low fracture toughness [31]. In addition, visible cracks were gradually showing on the coating surface during curing period, which extremely affected the adhesive behavior (discussed in Section 2.3). In contrast, no visible cracks in the coating #1 surface (Figure 5B), demonstrating that the brittle behavior was justified by compositing CMS. Evidently, according to data from Figure 4, the 28 d compressive/flexural strength of sample *3#* = 6.83 which was much lower than that of sample #1 of 7.60, indicating that the CMS polymer chains improved the elasticity of structure.

### 2.3. Water Retention, Shrinkage, and Cracking

The water retention results are shown in Figure 6, an increase of water retention with increasing content of CMS was detected, meanwhile water loss rate significantly slowed down and a higher final water content was obtained. These results could be attributed to the presence of the multiple hydrophilic group (-CH2COONa, -OH) in glucopyranose unit of CMS that effectively fixed the residual water by hydrogen bond to hold a higher humidity in AAM paste, so that resulted in lower escape of water fragment.

The effect of CMS on the drying shrinkage of AAM samples is plotted in Figure 7. CMS reduced the drying shrinkage of samples significantly. Combining the results of Figure 6 and Figure 7, it was found that both the drying shrinkage and water retention showed the significant and almost similar change trend in the first 14 d, indicating water fragment escape strongly effected the shrinkage behavior of AAM coating. For these results, it may be explained that the low water fragment removal reduced tension stress in capillary pores and volume change degree of coating [19]. On the other hand, it also may be explained that the improvement of toughness resulted from the filling effect of CMS organic polymer [30].

Large shrinkage behavior of AAM definitely led to volume change and subsequently caused cracking [19]. The coatings appearance after being 28 d cured in the ambient environment (Figure 8) indicated that CMS significantly reduced the risk of cracking. Lots of visible cracks presented in the coating #1 surface, corresponding to the largest shrinkage. Such a cracking problem will cause peeling and subsequently affect its mechanical strength. In contrast, it can be noticed that the cracking degree reduced with the increasing CMS. Though cracks still exhibited on the surface of coating #2, the cracked area, quantity, and width relatively reduced. Obviously, surface appearance of coating #3 was integrated and no visible cracks were observed while unfortunately showing small voids, implying that the cracking risk still presented. A previous study investigated the effect of MgO-based expansion and polypropylene fiber on the cracking resistance of AAM coating but found their effect to be very limited [32], because the AAM coating usually is a relatively thin layer, the water definitely quickly removed from the coating body after being brushed on the substrates, and then resulting in the large drying shrinkage during hardening. In this study, the addition of CMS gel could hold a high water content in the AAM paste during the solidifying process, which in turn reduced shrinkage and cracking. Furthermore, our previous study found that the padding effect of polycondensed aluminum phosphate in pore and micro-cracks can effectively control the cracking problem of the AAM coating [23]. For the shrinkage reduction results in here, it also can be attributed to the filling effect of the CMS binder, implying the elastic CMS organic gel bonded into the coating structural defects, and thus the thin coating may be able to carry tension stress before cracking.

### 2.4. Microstructure Analysis

The morphologies of the sample after being cured 28 d are shown in Figure 9. As expected, a significant change in microstructure with increasing CMS was clearly observed. The structure of plain AAM (Figure 9A) was much denser and smoother than that of the sample composited with CMS, corresponding to high mechanical strength. Meanwhile, due to the brittle behavior, distinct voids across in the AAM matrix were produced during the polishing process. However, it can be seen in sample #2 and #3 that compositing CMS considerably controlled cracking behavior. The continuous distributed CMS/AAM gel structure was detected in sample #3 (Figure 9C), implying the fracture toughness was improved. In addition, there is an obvious increase in coarse was observed in the sample #3, which correlated with the observed reduction of mechanical strength and shrinkage in this specimen. On the other hand, the CMS/AAM structure in microstructural homogeneity provided a strong reasoning for the enhancement in rheological behavior, as described in Section 2.1. Nevertheless, many incompletely dissolved CMS particles clearly presented in sample #4 (Figure 9 D) due to an unsuitable water/solid ratio, which resulted in stiff behavior.

Figure 10 presents the diffractograms of the CMS and coatings with curing time. The diffractogram of CMS (Figure 10A) was characterized by reflexes at diffraction angle in the positions 11.40, 15.02, 18.16, and 23.36 degree, which related to the CMS semi-crystalline structure [33]. Figure 10B shows that all the coatings displayed a typical AAM broad characteristic hump located in the 25–36 degree, meanwhile the characteristic peak of CMS completely disappeared in sample #3 after being cured for 1 d, suggesting that the damage of CMS semi-crystalline structure was quickly achieved during the suspension stage, and subsequently generated water soluble gel. This XRD result also reasonably explained the rheological altering mechanism of CMS in the AAM paste. In addition, the peak of samples had no change and showed similar amorphous structure even after being cured for 28 d, which implied that the addition of CMS had no effect on the alkali-activated reaction.

### 2.5. Applicable Test of Coating

Sample #3 was selected in the coating test due to appropriate workability and cracking resistance properties. In addition, in order to obtain the colorful appearance of coating, FeO(OH) was used as pigment powder for color matching here, and 20% wt quartz sand was additionally composited as filler in order to ensure the enough cracking resistance capacity. The fresh coating after mixing was horizontally coated on calcium silicate plates by using a scraper, then a template roller was used to form the flat surface into multiple-dimension layer structure. The coating can be coated well without discontinuity, sag and stiffness during the scraping process as can be seen in Figure 11A, and the high plasticity and thixotropy allowed the coating surface to be formed into complex layer structure with desired aesthetic impression (Figure 11B–D).

## 3. Materials and Methods

### 3.1. Materials

The primary raw materials used in this study were provided by local suppliers in China. Ground granulated blast furnace slag (GGBS) was obtained from Beihai Chengde Steel Company (Beihai, China). The chemical composition of slag is shown in Table 1 which was obtained from X-ray fluorescence (XRF). Its particles size distribution was obtained from a Malvern Mastersizer 3000 laser granulometry measurement (Malvern Instruments Ltd., Malvern, U.K.) and the results are shown in Figure 12. The mean particle size of slag (*d_50_* = 8.48 μm) used in this study was very similar to that of other research. The solid powder Na-based-water-galss with 1.8 mudulus and 78 % solid content was used as alkaline activator, which was produced by Nanning Chunxu Chemical Company (Nanning, China). CMS (AR grade) was provided from Cool chemical science and technology Co., LTD (Beijing, China). CMS is a non-regular shape with a micrometer diameter as shown in Figure 13, and its chemical structure is illustrated in Figure 14. The presence of functional groups (-CH_2_COO^-^) and (OH^-^) yielded CMS with many unique properties, such as low gelatinization temperature, excellent flexibility and high paste storage stability. Quartz sand with a particle size of about 100 mesh was obtained from Lingfeng Chemical reagent Co.,LTD (Shanghai, China).

### 3.2. Sample Preparations

The mix proportion of the coatings is shown in Table 2, *water/solid* = 0.40. The coating was manufactured by using powder technology, slag, water glass, and CMS which were mixed together in a blender mixer (Droide Company, Shanghai, China), and then the well dispersed powders were used as the one-part drying powder coating. To prepare coating, water was poured into the powder coating on the above and was dispensed to homogenous paste using an electric mixer with 100 rmp, 300 rmp, and 100 rmp for 1 min duration in each mixing process.

### 3.3. Experimental Procedures

#### 3.3.1. Rheology and Workability

The rheological behavior of AAM coating was determined by Anton Paar MCR-301 advanced rotation rheology equipment (Anton Paar Instruments Ltd., Graz, Austria)with a double gap cylinder cell, and analytical Start Rheoplus software (Rheopuls, version 2.0, Anton Paar:2009) was used for data calculation. The AAM coating was introduced into the measurement system after mixing. The rheological measurements were started after a period of rest of 60 s. The paste properties were evaluated from the flow curves determined for increasing and decreasing values within a shear stress range from 0.01–100 s^−1^. For this test, a ramp profile was set up with an upward ramp and a downward ramp using a step wise method. Pre-shearing with a shear rate of 100 s^−1^ was performed for 60 s, then the speed was ramped to 0.01–10 s^−1^, 10–50 s^−1^, 50–90 s^−1^ and 90–100 s^−1^ for a 20 s duration in each testing period, then sheared at 100 1/s for 20 s, and then ramped down to 100–90 s^−1^, 90–50 s^−1^, 50–10 s^−1^ and 10–0.01 s^−1^ for a 20 s duration in each testing period. The measurement was repeated six times at 5 min (0 h), 1, 2, 3 and 4 h, and each sample was tested three times. The thixotropy was calculated from the area between the increasing curve of 0.01–100 s^−1^ and decreasing curve of 100–0.01 s^−1^. The yield stress and plastic viscosity were calculated from the decreasing branch of 100–0.01 s^−1^ using the Bingham (1) and Herschel-Bulkley (2) [22] model described as:*τ* = τ_0_ + μ_p_ γ(1)
*τ* = τ_0_ + K γ^p^(2)
where *τ_0_* (Pa) = yield stress, *μ_p_* (Pa s) = plastic viscosity, *γ* (1/s) = shear rate, *K* (Pa.s^n^)= consistency coefficient, *P* = flow index.

#### 3.3.2. Mechanical Strength

The fresh AAM were poured and sealed in a mold with dimension 40 mm × 40 mm × 160 mm, then cured at 25 °C, *RH* = 90 %. After curing, the mechanical strength of the AAM blocks was tested using a DNS100 universal testing machine. Flexural strength of the AAM blocks was conducted using a three-point bending configuration in a displacement controlled mode at a displacement rate of 0.50 mm/min. After the flexural strength test, the sample was used to test compressive strength using a fixture with dimension of 40 mm × 40 mm × 40 mm at a displacement rate of 0.50 mm/min. The adhesive strength was tested by pull-off method. Fresh coatings were brushed on the concrete substrates and cured at 25 °C, *RH* = 90 % for 28 d prior to test, and the adhesive strength was obtained by the mean value of three times of each sample.

#### 3.3.3. Water Retention, Shrinkage and Cracking

The homogenous paste was poured in a plastic beaker, weighed the initial total mass and noted as m_1_, the samples were being cured at 25 °C, *RH* = 55%, then weighed the samples mass again and marked as m_2_, recorded their mass lost up to 28 d. Water retention rate testing was calculated by equation (3) and (4) described as:Initial water mass = m_1_ × 0.2857(3)
where 0.2857 = m(water)/[m(water)+m(slag)+m(water glass)], which calculated from Table 2
*Water retention rate* = 1 − [(m_1_ − m_2_) / (m_1_ × 0.2857)](4)

Drying shrinkage test was performed on a 25 mm × 25 mm × 280 mm specimen using a bench comparator with an accuracy of 0.001 mm to determine the linearity of the prismatic specimens along the longitudinal axes. The coating was poured in the mold and carefully demolded after being cured at 25 °C, *RH* = 90 % for 24 h. The initial longitudinal length was recorded immediately after being demolded. Then the samples were cured at 25 °C, *RH* = 55%, and tested the length change in triplicate for each sample for up to 28 d.

Fresh coatings were brushed on the concrete substrates and cured at 25 °C, *RH* = 55 % for 28 d, then the cracking behavior was recorded by a camera (Iphone-8 Plus), and was treated by using grayscale analysis methodology of Photoshop.

#### 3.3.4. Microstructure Analysis

The XRD characterization was conducted on a Rigaku MiniFlex 600 instrument with Ni-filtered Cu (K<alpha>) radiation. The instrument was operated at 40 kV and 15 mA, with a dwell time of 3 s, 2θ range of 5 to 70° and step size of 0.020°. SEM device was used to analyze the fragment morphology of the AAM sample with a 3400N device (Hitachi Limited Company, Tokyo, Japan) under an acceleration voltage of 15 kV condition.

## 4. Conclusions

In this study, a one-part plastic formable novel AAM coating has been successfully developed by compositing water soluble starch CMS which can be used for decorating coatings in building construction to obtain various graceful and artistic layer structure. Such organic/inorganic hybrid coating possessed high plasticity, stable workability, high adhesive strength, and volume stability. CMS could effectively enhance the yield stress, plastic viscosity, and thixotropy of the coating, imparting the prepared coating to be freely formed into complex layer structures. The rheological property improvement resulted from the water molecule immobilization by hydrogen effect of CMS and its gel intertwined between slag particles. Although the rheological parameters increased with time due to copolymerization, the optimum coating composited with 15.40 wt% CMS showed stable workability even after 4 h reaction. CMS reduced the mechanical strengths of coating, but the strengths still met the requirements of relevant standards. The CMS/AAM formation structure exhibited in microstructural homogeneity, resulting in strong improvement of volume stability and cracking resistance ability. The experiment results on the above suggested that the CMS is feasible to be used in an AAM system to remedy its inorganic shortages so that the CMS/AAM hybrid composite assembled both the feature of high strength and durability from AAM as well as excellent plasticity from organic polymer, which could be beneficially used as a decorative coating.

## Figures and Tables

**Figure 1 molecules-25-00844-f001:**
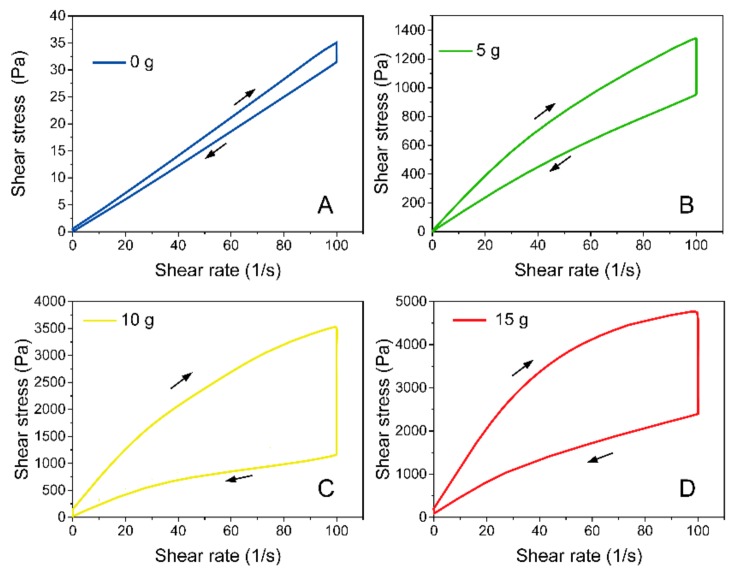
Flow curves of AAM coating with increasing CMS content. (A) plain AAM, (B) AAM composited with 5 g CMS, (C) AAM composited with 10 g CMS, (D) AAM composited with 15 g CMS.

**Figure 2 molecules-25-00844-f002:**
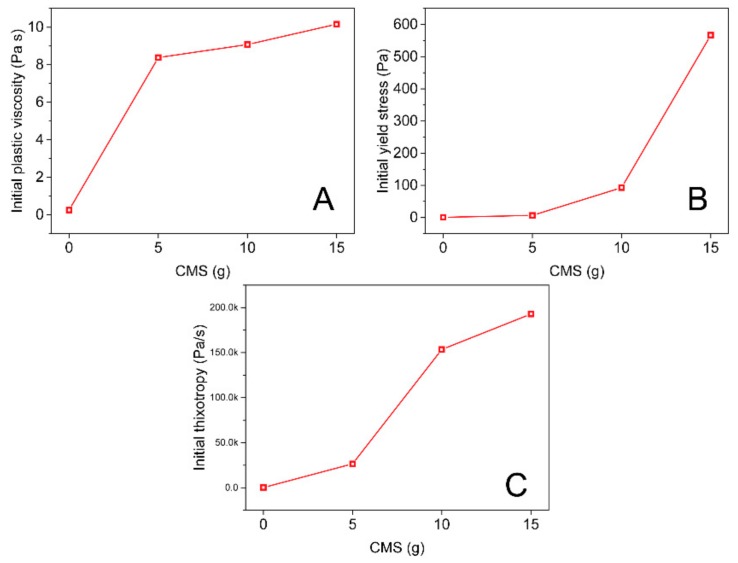
Effect of CMS on the initial plastic viscosity (**A**), yield stress (**B**) and thixotropy (**C**).

**Figure 3 molecules-25-00844-f003:**
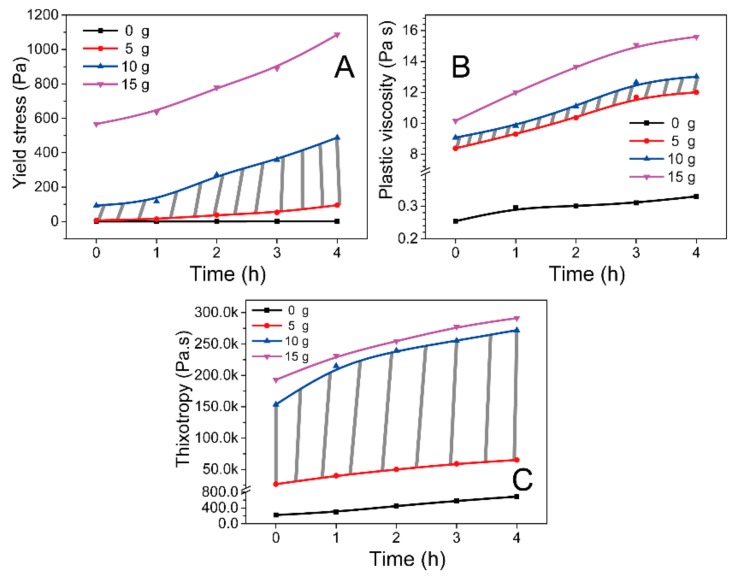
Effect of CMS on the evolution of yield stress (**A**), plastic viscosity (**B**) and thixotropy (**C**) with working time.

**Figure 4 molecules-25-00844-f004:**
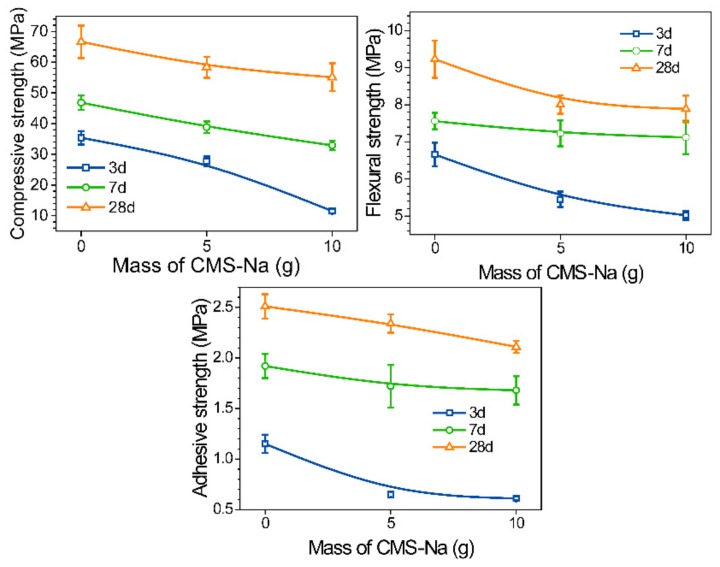
Effect of CMS on the compressive (**A**), flexural (**B**) and adhesive (**C**) strength of AAM.

**Figure 5 molecules-25-00844-f005:**
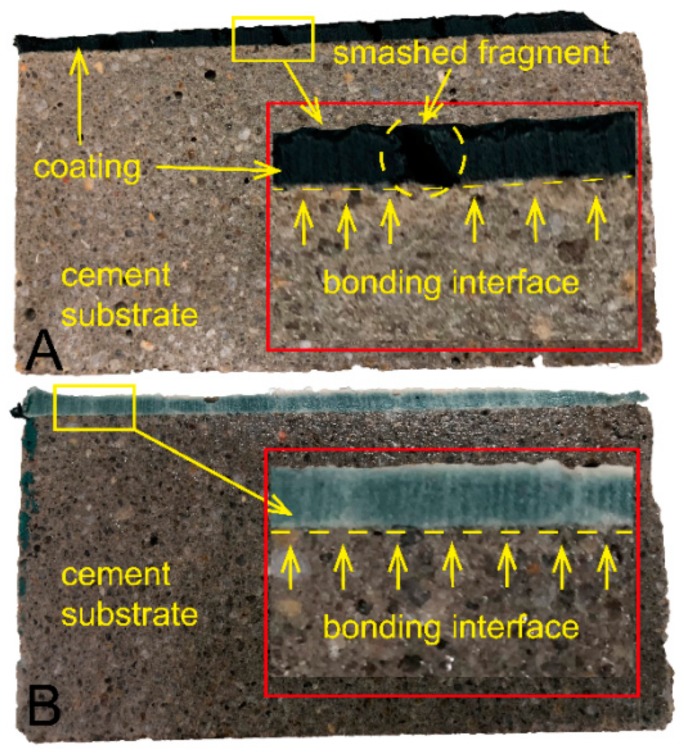
Adhesive characteristics between AAM coating and cement mortar substrate, (**A**) sample 1#; (**B**) sample 3#.

**Figure 6 molecules-25-00844-f006:**
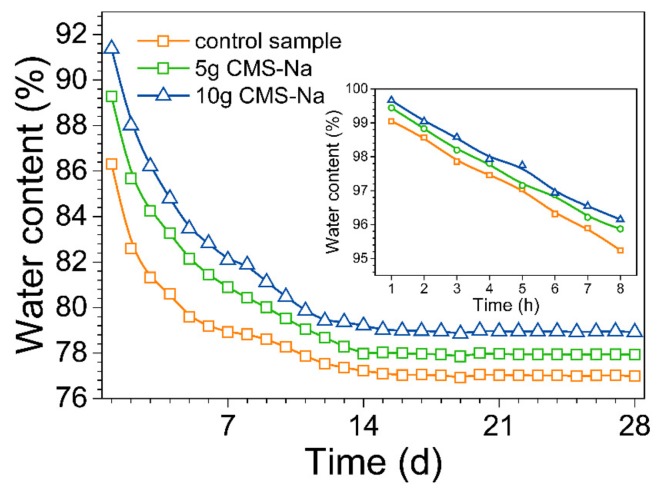
Effect of CMS on the water retention of AAM paste with curing time evolution.

**Figure 7 molecules-25-00844-f007:**
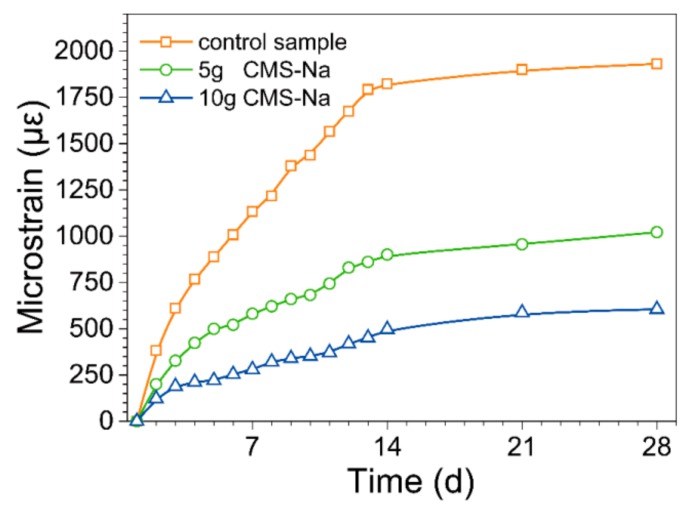
Effect of CMS on the drying shrinkage of AAM paste with curing time evolution.

**Figure 8 molecules-25-00844-f008:**
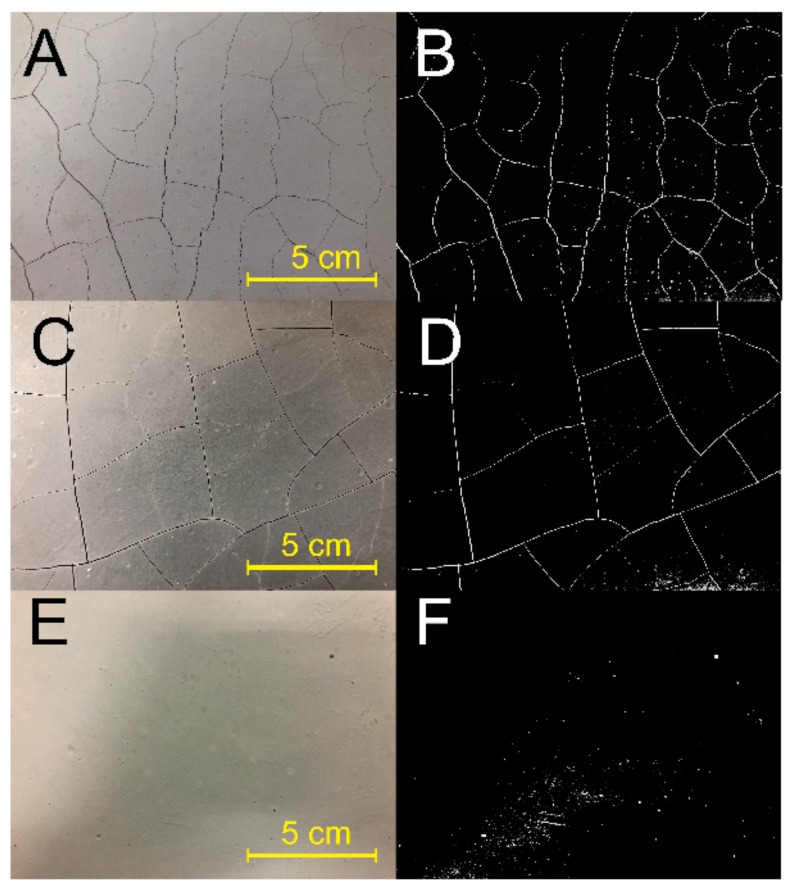
Coating appearance after being curing at ambient environment for 28 d. (**A**)\(**B**) sample 1#, (**C**)\(**D**) sample 2#, (**E**)\(**F**) sample 3#.

**Figure 9 molecules-25-00844-f009:**
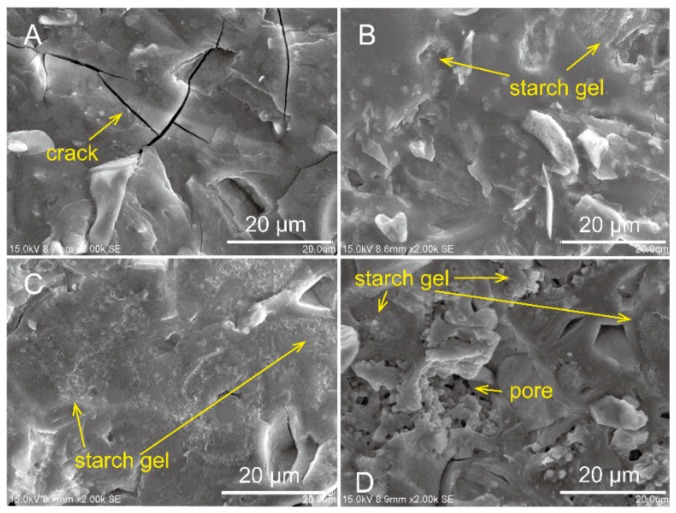
SEM image of AAM paste with various composition of CMS, **A** sample 1#, **B** sample 2#, **C** sample 3# and **D** sample #4.

**Figure 10 molecules-25-00844-f010:**
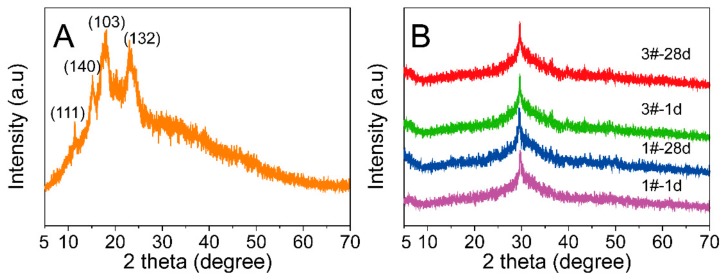
XRD patterns of CMS (**A**) and AAM coatings with different content CMS at time evolution (**B**).

**Figure 11 molecules-25-00844-f011:**
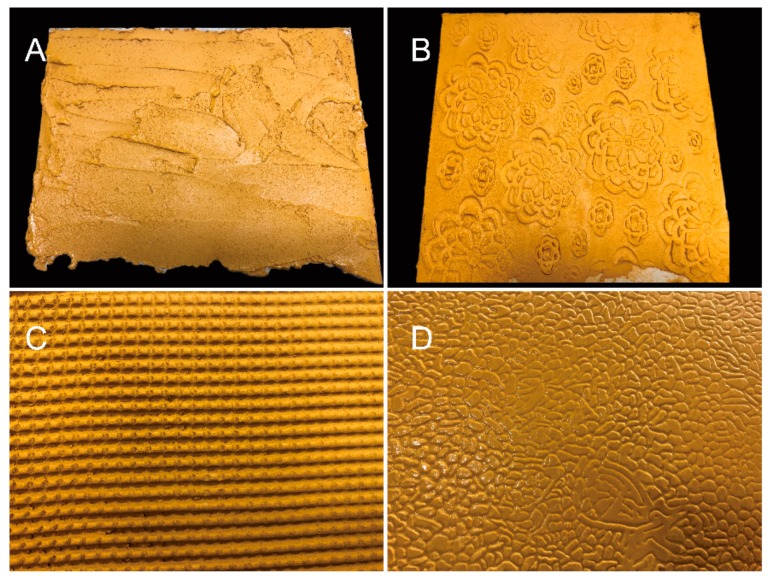
Picture to illustrate the applicable test of coating. (**A**) coating after being scraping, (**B**, **C**, **D**) multiple-dimension-coating-layer.

**Figure 12 molecules-25-00844-f012:**
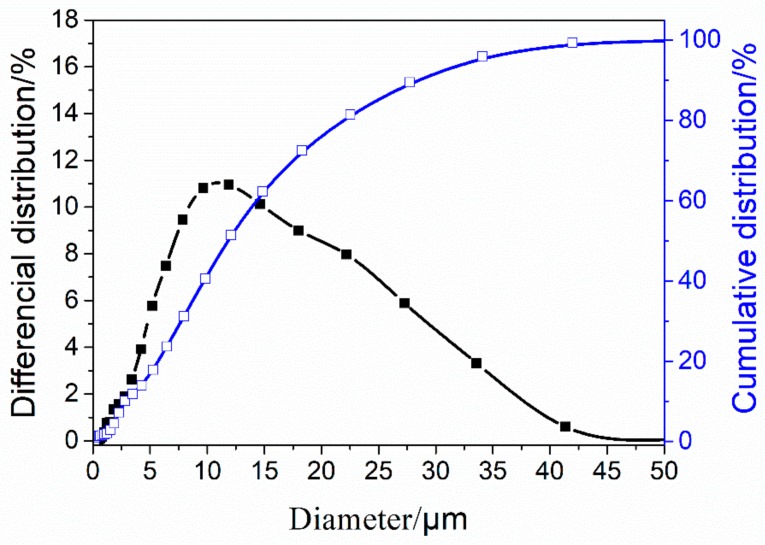
Particle size distribution of slag used as AAM precursor.

**Figure 13 molecules-25-00844-f013:**
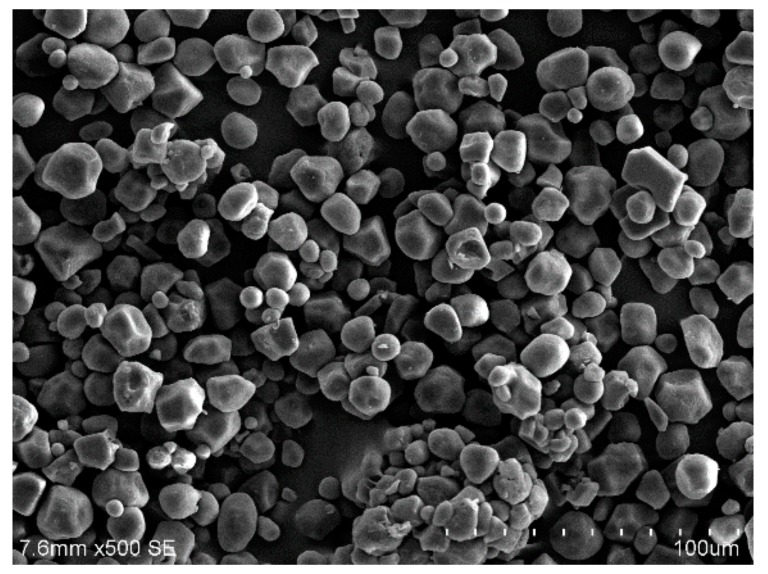
Scanning electron microscope (SEM) image of CMS particles.

**Figure 14 molecules-25-00844-f014:**
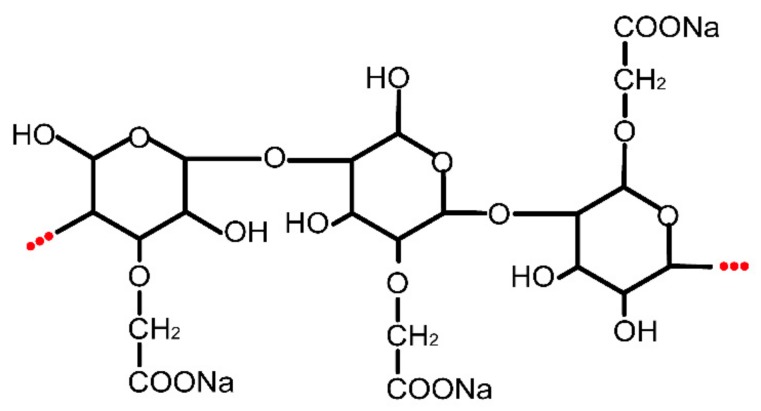
Chemical structure of CMS used as a viscosity modifying agent in AAM.

**Table 1 molecules-25-00844-t001:** Chemical composition of slag (mass %).

Composition	CaO	SiO_2_	Al_2_O_3_	MgO	SO_3_	TiO_2_	MnO	Fe_2_O_3_	K_2_O	Na_2_O	LOI^a^
Content	47.39	29.73	11.89	5.99	1.79	0.95	0.65	0.49	0.47	0.30	0.35

^a^ Loss on ignition.

**Table 2 molecules-25-00844-t002:** Coating formulation.

No.	Slag (g)	Water Glass (g)	CMS (g)	Distill Water (g)	CMS (%)
1#	50.00	15.00	0.00	26.00	0.00
2#	45.00	15.00	5.00	26.00	7.70
3#	40.00	15.00	10.00	26.00	15.40
4#	35.00	15.00	15.00	26.00	23.10

Note: *CMS (%)* = 100%×[CMS/(slag+water glass+CMS)].

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
