# Peer review of "One-Part Plastic Formable Inorganic Coating Obtain from Alkali-Activated Slag /Starch(CMS) Hybrid Composites"

_molecules, 2020, doi:10.3390/molecules25040844_

Round 1

Reviewer 1 Report

see attached file

Author Response

Thank you for review our manuscript. These comments are all valuable and very helpful for revising and improving our paper, making our research more meaningful. We have responded to the questions raised by the reviewers. The paper has been enriched further according to the revised proposal (the change in the text are in red font).

Reviewer 1#:

The scientific merit of the paper is rather limited, but the described new material might find practical application. Therefore, I recommend publication of the manuscript, however after serious corrections. First of all description of the preparation of a new material makes the sense only if this material presents some advantage over the existing ones. Those advantages might be: lower price, wider application, better accessibility of the raw materials, etc. That aspect is not sufficiently well discussed in the manuscript.

Reply: Thanks for the reviewer’s suggestions. We revised the introduction, and did the descriptions of the advantages of AAM coating by comparing with other existing traditional coating. These revisions were marked in red font in the revision manuscript in line of 54-72.

The manuscript is written in a lengthy manner. (a) For example it is not necessary to describe the history of the application of a geopolymer in such article. (b) Authors discuss exhaustively the observed viscosity increase and the appearance of thixotropic properties of the geopolymer after the addition of starch. It is trivial! What else can be expected? Starch is a very popular additive used to increase the viscosity of many preparation and its influence on the fluid behavior is well known. Author in many places repeat the information. (c) For example in page 5 (lines 155-171) author describes in details the data presented in figure 5, giving all numerical values with the accuracy of up to 4 significant digits. Such precise information is useless for the reader and the general trend is easy to recognize looking at the figure. (d) Conclusion section is too long and too detailed. Giving such information like plastic viscosity, yield strength and thixotropy of all preparation with accuracy of 4 significant digits is useless. Only general trends should be mentioned.

Reply: These are good advices to improve the logic and readability of this paper. The manuscript was revised as followed: (a) The history of the application of inorganic coating was deleted. (b) Those results described in too accuracy and useless sentences in the whole manuscript were rewritten, and more concise results description were given. In addition, instead of those exhaustive descriptions of the observed rheological results, we did detailed discussions on the influence of CMS in fresh AAM paste, and then discussed the relation of rheological performance with workability. (c) These too long and detail descriptions of the results were deleted or rewritten, we gave the general tends only and did a discussion for the observed results. (d) The conclusion section was rewritten; we deleted all the useless and too precise information like viscosity, yield stress and thixotropy, etc, giving the important finding conclusions of this work. These revisions on the above can be seen in red font in the revision manuscript.

Manuscript is written in rather bad English. There is a lot of grammatical and stylistic errors, some parts of the manuscript are simply difficult to understand. Below some examples are given, but they are only examples, the manuscript is full of such clumsy sentences:

(a) “…the viscosity significantly improved with the increasing CMS.” Viscosity is a measurable quantity-it may increase or decrease but what does it mean “Viscosity significantly improved?” It depends on what was expected.

(b) “Fig. 2 shows the trend of initial plastic viscosity, yield stress and thixotropy which fitted by the increasing of CMS.” What does it mean “fitted by the increasing cms.”? Experimental data may be fitted by an equation or function but not by “the increasing CMS”

(c) Author write: “Implicitly, rheology is one of the most important properties of the coating which related to the externally applied mechanical force and responded to workability” Rheology is a branch of science, not a property, and why imoplicitly?

(d) “A small scale application test was conducted in the lab to illustrate the application test of the prepared coating” It makes no sense.

Reply: We are so sorry for the clerical errors and poor English writing ability. However, we carefully checked the whole paper and revised those errors in the revision paper. And for reviewer’s comment, we reply as followed: (a) This sentence was pasted in the original manuscript in line 89-90, and now the sentence is deleted, as none viscosity value were presented in Fig. 1. In addition, we explained the reasons for the thixotropy increase after the addition of CMS in line 115-124 in revision paper. Because CMS produced colloidal interaction gel presented between slag particles, and CMS fixed part of the residual water by hydrogen bond, improving the cohesiveness of coating, thus effectively changed AAM visco-elastic behavior so that met the engineering requirements. (b) This sentence now is rewritten as “Fig. 2 shows the effect of CMS on the improvement degree of initial plastic viscosity, yield stress and thixotropy”, the fitted curves was deleted, and the improvement trends with the increasing CMS was given. (c) This sentence now is rewritten as “As AAM is a high reactive system and easily solidify, the evolution of coating flow behavior with time is one of the very important properties which relates to the operable time and workability during real application”, see in revision manuscript marked in red font in line 160-163. (d) This sentence was deleted.

Figures should be numbered consecutively as they appear in text. In the reviewed manuscript figures 9 and 10 appear in text after figure 1, figure 10 after figure 6 (lines 101 and 201)

Reply: We are so sorry for the clerical mistakes. Thanks for the reviewer’s reminder again. These errors was revised in the revision text, figures were consecutively numbered in the text.

Author write: “The thixotropy and viscosity increment in Fig. 1 B, C, D can be explained from that CMS quickly hydrolyzed in water to create entanglement and intertwining gel networks (evidently show in Fig. 9 and 10,….)” Figure 10 shows XRD patterns of CMS (A) and geopolymer coatings and does not support the above give conclusion.

Reply: The conclusion here is vague and was deleted.

Authors write units of different quantities in a wrong manner. For example for the viscosity they write the dimension “Pa.s”. No multiplication mark nor full stop is needed in the dimension, it should be “Pa s”.

Reply: Thanks for the reviewer’s reminder. The wrong units presented in whole paper were revised including figures, sentences and equations.

Numerical data should be given with the precision reflecting the accuracy of the measurement. For example authors gave the value of viscosity (line 113) as 10.161 Pa s. I do not believe they measured the viscosity with the accuracy of 5 significant digits. There are many such places in the manuscript.

Reply: Thanks for the reviewer’s reminder again. In this study, the results was obtained from an Anton Paar MCR-301 advanced rotation rheology equipment, and an analytical Start Rheoplus software 2.0 was used for the data recordation and calculation. This devise actually can provide data with 3 decimal places. We did not directly test the viscosity value in the test process; see in the paper of 3.3.1 section. The viscosity values in fact were calculated from equation of “shear stress divide to shear rate”, and we reserved 3 decimal places of the calculated data. This improper writing manner has been revised though out the whole paper.

Table 1. LOIa-no description is given.

Reply: Loss on ignition (LOI), the description was pasted below the Table 1. see in line 336.

Reviewer 2 Report

 The authors have investigated the applicability of geopolymer/starch hybrid composites for use as an inorganic coating material. The results presented in this work are valuable and will be of interest to the readers of this journal.

 Despite the quality of the experimental results, it is very unfortunate for the authors to have failed to differentiate alkali-activated slag from 'geopolymer'. The reaction product formed in the material used in this study is in fact more like C-N-A-S-H which would resemble to tobermorite (this is also supported by the XRD results in Fig. 10), while what we would typically expect from geopolymer is something that has zeolite framework. The material used in this work should never be referred to as geopolymer. They are completely irrelevant. I suggest the authors a major change in the introduction; the part revisiting works on geopolymer should all be removed, and the work pertaining to alkali-activated slag should be included. This would also mean that the samples should not be referred to as geopolymer (including in the title).

Author Response

Thank you for review our manuscript. These comments are all valuable and very helpful for revising and improving our paper, making our research more meaningful. We have responded to the questions raised by the reviewers. The paper has been enriched further according to the revised proposal (the change in the text are in red font).

Despite the quality of the experimental results, it is very unfortunate for the authors to have failed to differentiate alkali-activated slag from 'geopolymer'. The reaction product formed in the material used in this study is in fact more like C-N-A-S-H which would resemble to tobermorite (this is also supported by the XRD results in Fig. 10), while what we would typically expect from geopolymer is something that has zeolite framework. The material used in this work should never be referred to as geopolymer. They are completely irrelevant. I suggest the authors a major change in the introduction; the part revisiting works on geopolymer should all be removed, and the work pertaining to alkali-activated slag should be included. This would also mean that the samples should not be referred to as geopolymer (including in the title).

Reply: Thanks for your suggestions. We did a major change in the introduction, all “geopolymer” in the paper were replaced by “Alkali-activated slag” or “alkali-activated materials”, giving the description of alkali-activated materials in line 54-72 of revision manuscript.

Round 2

Reviewer 2 Report

The authors have well addressed my concerns. One last concern is that:

Lines 87-88 Page 2: alkali-activated slag will definitely have a 'simple Si-O-Si structure', even though 'simple' does not well describe the actual structure. It is certainly not a 3D framework...

The authors should correct this, before it is published.

Author Response

Dear Editors and Reviewers:

Thank you again for your professional suggestions to improve the quality of our manuscript.

Reviewer 1#:

Alkali-activated slag will definitely have a 'simple Si-O-Si structure', even though 'simple' does not well describe the actual structure. It is certainly not a 3D framework...

The authors should correct this, before it is published.

Reply: The description of gel in AAM binders was rewritten in lines51-52, “AAM binders have nanostructural amorphous or poorly-crystalline gel of C-(N)-A-S-H/N-A-S(-H)”.